# Transcriptome Analysis Reveals the Effect of PdhR in *Plesiomonas shigelloides*

**DOI:** 10.3390/ijms241914473

**Published:** 2023-09-23

**Authors:** Junxiang Yan, Bin Yang, Xinke Xue, Jinghao Li, Yuehua Li, Ang Li, Peng Ding, Boyang Cao

**Affiliations:** 1TEDA Institute of Biological Sciences and Biotechnology, Nankai University, Tianjin 300457, China; 2Key Laboratory of Molecular Microbiology and Technology of the Ministry of Education, Nankai University, Tianjin 300457, China; 3Tianjin Key Laboratory of Microbial Functional Genomics, TEDA College, Nankai University, Tianjin 300457, China; 4State Key Laboratory of Medicinal Chemical Biology, Nankai University, Tianjin 300353, China; 5College of Pharmacy Laboratory of Molecular Drug Research, Nankai University, Tianjin 300353, China

**Keywords:** *Plesiomonas shigelloides*, PdhR, RNA-seq, motility, T3SS, ArcA

## Abstract

The pyruvate dehydrogenase complex regulator (PdhR) was originally identified as a repressor of the *pdhR-aceEF-lpd* operon, which encodes the pyruvate dehydrogenase complex (PDHc) and PdhR itself. According to previous reports, PdhR plays a regulatory role in the physiological and metabolic pathways of bacteria. At present, the function of PdhR in *Plesiomonas shigelloides* is still poorly understood. In this study, RNA sequencing (RNA-Seq) of the wild-type strain and the Δ*pdhR* mutant strains was performed for comparison to identify the PdhR-controlled pathways, revealing that PdhR regulates ~7.38% of the *P. shigelloides* transcriptome. We found that the deletion of *pdhR* resulted in the downregulation of practically all polar and lateral flagella genes in *P. shigelloides*; meanwhile, motility assay and transmission electron microscopy (TEM) confirmed that the Δ*pdhR* mutant was non-motile and lacked flagella. Moreover, the results of RNA-seq and quantitative Real-Time Polymerase Chain Reaction (qRT-PCR) showed that PdhR positively regulated the expression of the T3SS cluster, and the Δ*pdhR* mutant significantly reduced the ability of *P. shigelloides* to infect Caco-2 cells compared with the WT. Consistent with previous research, pyruvate-sensing PdhR directly binds to its promoter and inhibits *pdhR*-*aceEF*-*lpd* operon expression. In addition, we identified two additional downstream genes, *metR* and *nuoA*, that are directly negatively regulated by PdhR. Furthermore, we also demonstrated that ArcA was identified as being located upstream of *pdhR* and *lpdA* and directly negatively regulating their expression. Overall, we revealed the function and regulatory pathway of PdhR, which will allow for a more in-depth investigation into *P. shigelloides* pathogenicity as well as the complex regulatory network.

## 1. Introduction

The genus *Plesiomonas*, represented by a single species, *Plesiomonas shigelloides*, is a gram-negative opportunistic pathogen associated with gastrointestinal and extraintestinal diseases in humans, such as acute secretory gastroenteritis, an invasive shigellosis-like disease, and a cholera-like illness [1,2,3,4]. *P. shigelloides* can grow in anaerobic and aerobic conditions, and it’s important to control the expression of genes involved in biosynthetic pathways, nutrient absorption, macromolecule synthesis, and excretion systems [5].

A crucial component of the metabolic connection between glycolysis and the citric acid cycle is the pyruvate dehydrogenase (PDH) multienzyme complex [6], which consists of pyruvate dehydrogenase (E1p), dehydrolipoate acyltransferase (E2p), and dihydrolipoate dehydrogenase (E3) and catalyzes the NAD-linked oxidative decarboxylation of pyruvate and the concomitant formation of acetyl-CoA [7,8]. The PDH complex is encoded by the operon *pdhR*-*aceE*-*aceF*-*lpdA* [6]. The pyruvate dehydrogenase complex regulator (PdhR), a transcriptional regulator of the GntR family that acts as a self-regulatory transcriptional regulator for this operon, is encoded by the *pdhR* gene [7,9]. The pyruvate-sensing PdhR represses the operon *pdhR*-*aceE*-*aceF*-*lpdA*, and pyruvate causes it to be derepressed [10]. The terminal product of glycolysis, pyruvate, is crucial in connecting a variety of metabolic pathways [11,12]. By using DNase I footprinting, the *pdh* operator was identified to be a region of hyphenated dyad symmetry, +11AATTGGTaagACCAATT+27, located immediately downstream of the transcript start site [7]. PdhR suppressed P*pdh* transcription in vitro more than 1000-fold, and PdhR represses transcription by binding to an operator site centered at +19 such that effective binding of RNA polymerase is prevented [7]. It has been reported that the operon *pdhR*-*aceE*-*aceF*-*lpdA* contains two promoters, the upstream *pdh* and internal *lpd* promoters [13]. Kaleta C et al. suggested that PdhR also regulates *lipA*, which encodes the lipoate synthase; meanwhile, they predicted a set of five novel TF-target gene interactions in *E. coli*. One of them, the regulation of *lipA* by the transcriptional regulator PdhR, was validated experimentally [14]. However, Feng Y and Cronan JE reported in vivo and in vitro evidence that *lipA* is not controlled by PdhR and that the putative regulatory site deduced by the prior workers is nonfunctional and physiologically irrelevant [15]. Cunningham L et al. also found that PdhR was not bound to the *lpd* promoter [13]. Similarly, we analyzed and predicted the operon *pdhR*-*aceE*-*aceF*-*lpdA* in the genome of *P. shigelloides* and found that it also contained the *pdh* and *lpd* promoters. Consistent with the above reports, PdhR can bind its own promoter but not the *lpd* promoter.

Aside from the *pdhR* operon, the *yfiD* gene, which encodes a putative formate acetyltransferase that is induced at pyruvate or low pH, has been identified as the first PdhR regulatory target [16,17]. Ogasawara H et al. identified two novel targets of PdhR [10], *ndh*, encoding NADH dehydrogenase II, and *cyoABCDE*, encoding the cytochrome bo-type oxidase [18,19,20], both together forming the pathway of respiratory electron transport downstream from the PDH cycle. Furthermore, the *lldPRD* operon of *E. coli* is responsible for aerobic l-lactate metabolism, and it has been proposed that LldR and its homolog PdhR act as regulators of the operon [21]. Göhler AK et al. discovered that the *glcDEFGBA* operon (genes for glycolate utilization, malate synthase), as well as the *mraZW*-*ftsLI*-*murEF*-*mraY*-*murD-ftsW*-*murGC*-*ddlB*-*ftsQAZ*-*lpxC* transcription unit (genes for proteins involved in cell division), are controlled by PdhR [6]. The regulation of the *fecABCDE* operon (genes for ferric citrate transporter) by PdhR has also been used to describe a connection between central metabolism and iron transport [22]. Meanwhile, Anzai T et al. performed genomic SELEX (gSELEX) screening in vitro, and they hypothesize that PdhR is a bifunctional global regulator for control of a total of 16–23 targets, including some genes for the surrounding pyruvate-sensing cellular pathways in addition to the genes involved in central carbon metabolism. Additionally, they revealed PdhR’s involvement in the positive regulation of genes involved in fatty acid degradation and the negative regulation of genes involved in cell motility [23]. So far, almost all reports on PdhR, including those described above, have come from the studies of *E. coli*, and there are few reports on other strains. In this study, we found that PdhR positively regulates flagella synthesis and T3SS, thereby affecting the motility and virulence of *P. shigelloides.* Furthermore, we hypothesized that, in addition to *pdhR*, PdhR directly regulates two downstream genes, *metR* (which participates in controlling several genes involved in methionine biosynthesis) and *nuoA* (the first gene of the NADH-quinone oxidoreductase gene cluster (*nuoA*-*N*)).

The O_2_-sensing transcription factor FNR regulates the expression of the *pdhR*-*aceEF*-*lpdA* operon as well as certain other PdhR-regulated genes [24,25,26,27]. FNR and PdhR collaborate to regulate gene expression and optimize metabolism by integrating responses to an environmental (O_2_) and a metabolic (pyruvate) signal [28]. ArcA is initially shown to anaerobically inhibit the PDH and 2-oxoglutarate dehydrogenase (ODH) complexes as well as succinate dehydrogenase (SDH) in enzyme experiments using an *arcA* mutant [29]. In the case of the PDH complex, ArcA may regulate one or both of the two important promoters (P*pdh* and P*lpd*) [30]. There are also reports that show *aceE* and *aceF* are positively regulated by cAMP-CRP and dually regulated by FNR [31]. However, in this study, we confirmed that both *pdh* and *lpd* are directly negatively regulated by ArcA, while FNR and cAMP-CRP indirectly positively regulate the expression of the *pdhR*-*aceEF*-*lpdA* operon.

In summary, we demonstrated that PdhR directly negatively regulates the expression of *pdhR-aceEF*, *metR,* and *nuoA* and indirectly positively regulates the expression of flagellar genes and T3SS genes, thereby affecting the flagellar synthesis and virulence of *P. shigelloides*; meanwhile, *pdhR* is directly negatively regulated by ArcA and indirectly positively regulated by FNR and cAMP-CRP. Furthermore, this work not only described the role and function of PdhR in *P. shigelloides*, but it also demonstrated for the first time that PdhR has a regulatory influence on T3SS. These findings may serve as a link between the global regulatory proteins and the virulence factors in the pathogenesis of *P. shigelloides*.

## 2. Results

### 2.1. Phylogenetic Analysis of PdhR

In *P. shigelloides*, the pyruvate dehydrogenase complex regulator PdhR was composed of 777 bases and 258 amino acids, with a protein size of 29.39 KDa. A phylogenetic tree based on PdhR amino acid sequences was constructed using the neighbor-joining method and plotted by MEGA 6.0. Bootstrap analysis was carried out based on 1000 replicates. The comparison results showed that PdhR is conserved in all the selected bacteria and is more closely related to *Escherichia coli* and *Salmonella enterica* than *Vibrio cholerae* and *Vibrio Parahaemolyticus* (Figure 1). As previously stated, almost all the reports on PdhR come from the study of *E. coli*, and PdhR plays an important role in regulating the physiological metabolism of *E. coli*. In addition to acting as a self-regulatory transcriptional regulator for the operon *pdhR*-*aceEF*-*lpdA*, PdhR also regulates formate acetyltransferase, NADH dehydrogenase II, cytochrome bo-type oxidase, l-lactate metabolism, glycolate utilization, malate synthase, cell division, ferric citrate transporter, and fatty acid degradation in *E. coli*. Therefore, the overview of studies on PdhR in *E. coli*, the construction of a PdhR phylogenetic tree, and the exploration of the function of PdhR in *P. shigelloides* in this study will lay the foundation for the diversified functional study of PdhR in different strains.

### 2.2. Transcriptome Sequencing Revealed Gene Expression Related to PdhR of the P. shigelloides

In this study, the transcriptome profiles of the WT and Δ*pdhR* strains were analyzed using RNA-seq to reveal the effect of PdhR in *P. shigelloides*. The RNA-seq results suggested that PdhR regulates approximately 7.38% of the *P. shigelloides* transcriptome: a total of 236 DEGs in the Δ*pdhR* strain were identified in comparison with the WT strain, including 190 downregulated genes and 46 upregulated genes (Figure 2A and Appendix A). Kyoto Encyclopedia of Genes and Genomes (KEGG) signaling pathway analysis showed that the downregulated genes were involved in flagellar assembly (*flgP*/*O*/*T*, *flgA*/*M*/*N*, *flgB-L*, *flaC*/*G*/*H*/*I*, *fliE-R*, *flhA*/*F*/*G-fliA*, *fliM*/*N*/*P*/*Q*/*R_L_*, *fliE-J_L_*, *flgB-L_L_*, *fliC*, *lafB*/*C*/*X*/*E*/*F*/*S*/*T*/*U*), bacterial chemotaxis (*cheV*/*R*, *motA*/*B*, *cheY*/*Z*/*A*/*B*/*W*), two-component system (*motA*/*B*, *cheV*/*R*, *cheY, cheA*, *lafT*, *fliA*, *hybC*, *ntrC*), bacterial secretion system (*iacP*, *sipA*/*D*/*C*/*B, spaT*/*P, invA*/*E*/*G*/*F, hilA, iagB, prgH*/*I*/*J*/*K, orgA*), microbial metabolism in diverse environments (*napF*/*D*/*A*/*A*/*G*/*H*/*B*/*C*, *cysD*/*G1*/*H*/*I*/*J*/*C*/*N*, *mocC*, *idhA*, *hybC*), sulfur metabolism (*cysD*/*G1*/*H*/*I*/*J*/*C*/*N*), folate biosynthesis (*moaA*/*B*/*C*/*D*/*E*), and ABC transporters (*modA*/*B*/*C*) (Figure 2B). Among the upregulated DEGs of KEGG analysis were genes responsible for the biosynthesis of secondary metabolites (*sdhC*/*D*/*A*/*B*, *aceE*/*F-lpdA*, *suhB*) and citrate cycle (TCA cycle) (*sdhC*/*D*/*A*/*B*, *aceE*/*F-lpdA*) (Figure 2C). Additionally, 12 upregulated and 13 downregulated DEGs in the transcriptome profiles were selected, respectively, for validation using qRT-PCR in the WT, Δ*pdhR*, and Δ*pdhR*/*pdhR*^+^ strains. The results of qRT-PCR were consistent with RNA-seq analysis (Figure 2D–G), indicating the reliability of the RNA-seq.

### 2.3. PdhR Influences Motility and Flagellar Synthesis by Positively Regulating the Expression of Flagellar Genes in P. shigelloides

We realized that PdhR may have an influence on *P. shigelloides* motility and flagellar synthesis after observing a high number of downregulated polar and lateral flagellar genes in transcriptome profiles (Figure 3A,C). Meanwhile, qRT-PCR was performed to validate the significant downregulation of 11 polar and 9 lateral flagellar genes (Figure 3B,D) in RNA-seq. In addition, 4 polar and 5 lateral flagellar genes were also chosen to construct promoter-lux fusions in the Δ*pdhR* mutant and WT strains to confirm the results of qRT-PCR and RNA-seq (Figure 3E). These results suggest that PdhR is a positive regulator of flagellar gene expression in *P. shigelloides*. Subsequently, we observed the migration of the WT, Δ*pdhR*, and Δ*pdhR*/*pdhR*^+^ strains in swimming agar plates, and validated the positive effect of PdhR on the motility of *P. shigelloides* (Figure 3F). Furthermore, the flagella produced by the WT, Δ*pdhR*, and Δ*pdhR*/*pdhR*^+^ strains were observed using TEM, which indicated that a lack of PdhR influences the flagellar synthesis in *P. shigelloides*, consistent with the motility assay (Figure 3G). All of the aforementioned findings showed that PdhR influences motility and flagellar synthesis by positively regulating the expression of flagellar genes in *P. shigelloides*.

### 2.4. PdhR Promotes P. shigelloides’ Ability to Infect Caco-2 Cells by Positively Regulating T3SS Expression

Previously, there was no pertinent research that mentioned PdhR’s impact on bacterial pathogenicity. However, we discovered that the Δ*pdhR* mutant’s transcriptome profile showed a downregulation of the T3SS cluster (Figure 4A), which implies that PdhR may also be a virulence regulator in *P. shigelloides*. In the subsequent studies, we chose 14 virulence genes from the T3SS cluster for qRT-PCR verification; the results were consistent with RNA-seq, and PdhR positively controlled T3SS cluster expression (Figure 4B). At the same time, before phenotyping the WT, Δ*pdhR*, and Δ*pdhR*/*pdhR*^+^ strains by invasion assay to confirm whether PdhR affects *P. shigelloides’* ability to infect Caco-2 cells, we utilized LB liquid medium, M9 medium, and DMEM to verify the effect of PdhR on the growth of *P. shigelloides*. The data showed that *P. shigelloides*’ overall growth and reproduction in the lag and log phases were not significantly impacted by PdhR (Figure 4C–E). Subsequently, invasion experiments of the WT, Δ*pdhR*, and Δ*pdhR*/*pdhR*^+^ strains revealed that PdhR considerably improves *P. shigelloides*’ capacity to infect Caco-2 cells (Figure 4F).

### 2.5. Pyruvate-Sensing PdhR Directly Negatively Regulates the Expression of pdhR-aceEF, metR, and nuoA

Based on previous research and the PdhR binding sites (Appendix A) published on the PRODORIC website (https://www.prodoric.de/matrix/MX000157.html (accessed on 12 March 2022)), we analyzed the promoter regions of all DEGs in RNA-seq, finally identifying potential PdhR binding sites in the promoters of five genes, namely *pdhR*, n*apF*, *metR*, *nuoA*, and *tfoX* (Appendix A). Then the EMSAs were performed to confirm a direct interaction between PdhR and the abovementioned promoter regions. The data showed that the complex of PdhR protein and DNA was observed when incubated with *pdhR*, *metR,* and *nuoA* promoter fragments (Figure 5A,B); however, PdhR was not bound to the promoters of n*apF, tfoX* (Figure 5C), or 16S rDNA as a negative control (Figure 5D). The result that PdhR still binds its own promoter in *P. shigelloides* is consistent with previous studies in other strains; however, we found the additional downstream genes *metR* and *nuoA* of PdhR, which have never been demonstrated before. In addition, we found that the operon *pdhR*-*aceE*-*aceF*-*lpdA* also contained *pdh* and *lpd* promoters in the genome of *P. shigelloides*, and PdhR was not bound to the *lpd* promoter (Appendix A). In this study, RNA-seq and qRT-PCR have verified PdhR as a repressor regulating the expression of the operon *pdhR*-*aceE*-*aceF*-*lpdA*, *metR,* and *nuoA* (Figure 2D–G). To further confirm our results, we constructed the *pdhR*, *metR,* and *nuoA* promoter-lux fusions in the Δ*pdhR* mutant and WT strains for the lux assay and the pBAD33-*pdhR*-3×Flag recombinant plasmids in the WT strains for the ChIP-qPCR assay. The results showed that the expression levels of *pdhR*, *metR,* and *nuoA* promoter-*lux* fusions in the Δ*pdhR* mutant (Figure 5E) were consistent with RNA-seq and qRT-PCR. Furthermore, the PdhR proteins were enriched at the *pdhR*, *metR,* and *nuoA* promoters, as shown in Figure 5F, with an 18.9-, 6.8-, and 11.7-fold higher signal in the ChIP samples than in the mock-ChIP samples (Figure 5F). Furthermore, as previously reported, we found that PdhR was able to sense the level of pyruvate, and that pyruvate could relieve the downstream inhibitory effect of PdhR in *P. shigelloides*. The expression levels of the operon *pdhR*-*aceE*-*aceF*-*lpdA*, *metR*, and *nuoA* were all elevated on either LB medium or M9 medium with 0.1% pyruvate (Figure 5G,H), consistent with the effect of *pdhR* deletion. These results described above confirm that pyruvate-sensing PdhR directly negatively regulates the expression of *pdhR-aceEF*, *metR,* and *nuoA*.

### 2.6. pdhR Is Directly Negatively Regulated by ArcA and Indirectly Positively Regulated by FNR and CRP

Currently, RegulonDB (https://regulondb.ccg.unam.mx/ (accessed on 6 February 2022)) has published six regulators that may interact with the promoter of *pdhR*, namely PdhR, FNR, ArcA, cAMP-CRP, Cra, and BtsR, whereas RegPrecise (https://regprecise.lbl.gov/ (accessed on 8 February 2022)) has published five regulators that may interact with the promoter of *pdhR*, namely PdhR, FNR, cAMP-CRP, FruR, and NarP. Additionally, using PRODORIC to predict the promoter of *pdhR*, we found that PdhR, FNR, ArcA, and cAMP-CRP may interact with the promoter of *pdhR* in *P. shigelloides* (Appendix A). In the abovementioned experiments, the combination of PdhR and its own promoter has been confirmed (Figure 5A). We subsequently verified direct interactions between FNR, ArcA, cAMP-CRP, and the promoter of *pdhR* by EMSAs, while we also verified direct interactions between FNR, ArcA, cAMP-CRP, and the promoter of *lpdA* in the operon *pdhR*-*aceE*-*aceF*-*lpdA*. The results indicated that phosphorylated ArcA (ArcA-P) can directly bind the promoters of *pdhR* and *lpdA*, while FNR and cAMP-CRP do not (Figure 6A–E). In order to demonstrate that direct ArcA binding to the *pdhR* and *lpdA* promoters is not non-specific, we also used the EMSAs for unphosphorylated ArcA (ArcA-P (-)) and the *lpdA* promoter (Figure 5B), as well as phosphorylated ArcA and 16S rDNA (Appendix A), as negative controls. Furthermore, the ChIP-qPCR assay revealed that ArcA proteins were enriched at the *pdhR* and *lpdA* promoters, with an 8.7- and 7.6-fold greater signal in the ChIP samples compared to the mock-ChIP samples, and further confirmed the direct regulatory relationship between ArcA and the *pdhR* and *lpdA* promoters (Figure 6F). We performed qRT-PCR and constructed *pdhR* and *lpdA* promoter-lux fusions for the lux assay in the Δ*arcA*, Δ*fnr*, and Δ*crp* mutant and WT strains to further clarify the positive or negative regulatory relationship between the three regulatory factors of ArcA, FNR, and cAMP-CRP and the operon *pdhR*-*aceE*-*aceF*-*lpdA*. In addition, both qRT-PCR and lux assays in Δ*arcA and* Δ*fnr* mutant strains were carried out under anaerobic conditions since ArcA and FNR play a regulatory role in this environment. The preceding experimental results demonstrated that ArcA negatively regulated the operon *pdhR*-*aceE*-*aceF*-*lpdA*, while FNR and cAMP-CRP positively regulated it (Figure 6G–J). Finally, our results demonstrated that the promoters of *pdhR* and *lpdA* in the operon *pdhR*-*aceE*-*aceF*-*lpdA* are directly negatively regulated by ArcA and indirectly positively regulated by FNR and cAMP-CRP.

### 2.7. The Potential Regulatory Pathways of PdhR in P. shigelloides

In the current investigation, we revealed the PdhR-controlled pathways and our proposal regarding the major regulatory pathways regulated by PdhR in *P. shigelloides* is outlined in Figure 7.

## 3. Discussion

The human pathogen *Plesiomonas shigelloides*, which causes intestinal infections and produces an inflammatory response, is often isolated from seafood, uncooked food, and contaminated water [32,33,34]. Therefore, research into *P. shigelloides*’ pathogenic mechanisms are necessary. In this work, the transcriptome profiles of WT and Δ*pdhR* strains were analyzed using RNA-seq to investigate the regulatory role of PdhR in *P. shigelloides*. The RNA-seq results suggested that PdhR regulates 236 DEGs, comprising 190 downregulated genes and 46 upregulated genes, of the *P. shigelloides* transcriptome. The downregulated genes in Δ*pdhR* strain transcriptome were mainly involved in the polar and lateral flagella for synthesis and assembly, T3SS for structural proteins and the effectors, periplasmic nitrate reductase (*napF*/*D*/*A*/*A*/*G*/*H*/*B*/*C*), assimilatory sulfate reduction (*cysD*/*G1*/*H*/*I*/*J*/*C*/*N*), molybdopterin biosynthesis (*moaA*/*B*/*C*/*D*/*E*), ABC-type transporter (*modA*/*B*/*C*), and hemolysin (*phlA*/*B*) (Figure 7). The upregulated genes were mainly involved in NADH-quinone oxidoreductase (*nuoA*/*B*/*C*/*D*/*E*/*F*/*G*/*H*/*I*/*J*/*K*/*L*/*M*/*N*), pyruvate dehydrogenase (*aceEF-lpd*), succinate dehydrogenase (*sdhC*/*D/A*/*B*), the methionine metabolism regulator (*metR*), and the long-chain fatty acid sensor (*psrA*) (Figure 7). Our report reveals, for the first time, the effect of PdhR as a regulator on *P. shigelloides*, which mostly includes positively regulating the expression of flagellar genes and TS33 genes to influence *P. shigelloides* motility and virulence. However, from the results of RNA-seq, PdhR also has a certain impact on *P. shigelloides*’ physiological metabolism, and we expect that future studies in this field will be explored.

Contrary to the findings of our study, previous research reported that PdhR represses flagellar synthesis and motility by regulating the *fliAZ* operon in *E. coli* [23]; our report revealed that the Δ*pdhR* mutant was non-motile and lacked flagella via motility assay and TEM, additionally, the results of RNA-seq, qRT-PCR, and lux assay all revealed that PdhR positively regulates the polar and lateral flagella genes in *P. shigelloides*. As for PdhR’s opposing influence on the regulation of motility between *E. coli* and *P. shigelloides*, we speculated that the main reason may be that the transcriptional regulatory factor PdhR may present diversified regulatory modes in different bacterial species. Such as the global transcription factor ArcA, which positively regulates the motility in *E. coli* and *S. Typhimurium* [35,36]; instead, in *V. cholerae*, ArcA negatively regulates its motility [37], while in *Serratia marcescens*, ArcA does not affect its motility at all [38].

Gram-negative bacteria are known to use a wide range of virulence factors to subvert eukaryotic cell physiological systems, with the type three-secretion system (T3SS) being one of the most important. The T3SS is a needle-like device that the bacterium employs to inject a varied collection of effector proteins straight into the cytoplasm of host cells, where they can disrupt the host cellular machinery for a variety of reasons [39]. In this study, we also found that PdhR positively regulates the expression of the whole T3SS gene cluster and that the ability of the Δ*pdhR* strain to infect Caco-2 cells was dramatically reduced when compared to WT. Since no previous studies have investigated the relationship between PdhR and the bacterial secretion system and bacterial pathogenicity, this is the first time we have shown that PdhR regulates the expression of bacterial T3SS and influences *P. shigelloides* virulence via the aid of the transcriptome and invasion assay. Even though we found that PdhR positively regulates *P. shigelloides* motility and virulence, the precise regulation mechanism of PdhR remains unknown to us. We did not locate the PdhR binding site in the promoter region of the flagellar or T3SS genes after analyzing the promoters of all DEGs; therefore, we expect to clarify the particular regulatory mechanism of PdhR in future studies.

The operon *pdhR*-*aceE*-*aceF*-*lpdA*, which contains *pdhR* and *lpdA* promoters, is known to encode the pyruvate dehydrogenase complex (PDHc) and PdhR [10,13]. Meanwhile, it is also accepted that PdhR inhibits the operon *pdhR*-*aceE*-*aceF*-*lpdA* and directly binds the promoter region of *pdhR*, but whether PdhR binds to the promoter of *lpdA* is controversial [13,14,15]. In this study, Softberry (http://www.softberry.com/ (accessed on 11 March 2022)) was used to predict the promoters of the four genes in the operon *pdhR*-*aceE*-*aceF*-*lpdA*, which found -10 and -35 regions exclusively in the *pdhR* and *lpdA* promoters (Appendix A). Consistent with earlier reports, the operon *pdhR*-*aceE*-*aceF*-*lpdA* in *P. shigelloides* occupies both *pdhR* and *lpdA* promoters; however, PdhR can only bind to its own promoter but not the *lpd* promoter.

In addition to *pdhR*, PdhR binding sites were also predicted in the promoter regions of n*apF*, *metR*, *nuoA*, and *tfoX*; however, EMSAs showed that PdhR binds to the promoters of *metR* and *nuoA*, but not the one of n*apF* or *tfoX.* Methionine is a unique sulfur-containing amino acid that plays an important role in biological protein synthesis and various cellular processes that play crucial roles in the pathogenesis of many microbial pathogens [40,41,42]. MetR participates in controlling several genes involved in methionine biosynthesis [43]. Homocysteine, an intermediate in the biosynthesis of methionine [44] binds to MetR and enhances the activity of some MetR-activated promoters (*metE* and *glyA*) by enhancing the affinity in DNA-binding sites [43,45]. But homocysteine is also able to decrease the activity of other promoters that are activated (*metH*, *metA*, and *hmp*) or repressed (*metR*) by MetR [43,44,46]. In addition, studies have revealed that MetR also regulates some other important cellular processes, including cell motility, H_2_O_2_ tolerance, heat tolerance, exopolysaccharide synthesis, and biofilm formation in *S. marcescens* [47]. While demonstrating that *metR*, the encoded protein acts as a regulator (MetR), as a downstream gene of PdhR, we also speculate whether MetR is involved in the regulatory mechanism of PdhR and will continue to investigate MetR’s effects in *P. shigelloides*.

The prokaryotic proton-translocating NADH-quinone oxidoreductase (NDH-1) is an L-shaped membrane-bound enzyme that contains 14 subunits (NuoA-NuoN or Nqo1-Nqo14) [48]. NDH-1 consists of two domains: the peripheral arm (NuoB, -C, -D, -E, -F, -G, and -I) and the membrane arm (NuoA, -H, -J, -K, -L, -M, and -N) [44]. In this study, we found that the 14 genes *nuoA*-*nuoN*, encoding NDH-1, are located on a gene cluster in the *P. shigelloides* genome (Appendix A) and that PdhR directly negatively regulates the expression of this gene cluster. Additionally, some of the regulating targets for PdhR previously described in other strains, such as *fecABCD* [22] and *cyoABCD* [10], do not exist in *P. shigelloides*. The other part is that PdhR binding sites are not found in promoter regions of the regulating targets in *P. shigelloides*, such as *yfiD* [16] (*pflB* in *P. shigelloides*), *ndh* [10] and *mraZW*-*ftsLI*-*murEF*-*mraY*-*murD*-*ftsW*-*murGC*- *ddlB*-*ftsQAZ*-*lpxC* [6]. Although the foregoing revealed that the regulatory role of PdhR in different bacterial species has certain differences, it also demonstrated PdhR’s importance and universality in the regulation of bacterial physiological metabolism.

The Arc two-component signal transduction system, comprising the kinase sensor ArcB and its cognate response regulator ArcA, is one of the mechanisms that enable bacteria to adapt to changing oxygen availability [49,50]. ArcA inhibits the expression of genes required for aerobic metabolism, energy generation, amino acid transport, and fatty acid transport Under anaerobic conditions [51]. In this work, we found that there were three binding sites of ArcA in the promoter region of *pdhR*, and we also demonstrated that ArcA directly negatively regulates *pdhR* expression by EMSAs, ChIP-qPCR, qRT-PCR, and the lux assay. However, we are not yet sure which of the three binding sites of ArcA in the promoter region of *pdhR* interacts with ArcA, which is what we need to verify next. Meanwhile, we also confirmed that ArcA also binds the promoter region of *lpdA* in *P. shigelloides*, which is consistent with previous reports [13]. In addition, we also found the binding sites of FNR and cAMP-CRP at the promoter of *pdhR.* Despite the fact that FNR and cAMP-CRP do not directly bind to the *pdhR* promoter, the results of the qRT-PCR and lux assay experiments showed that the deletion of either *fnr* or *crp* results in the downregulation of the operon *pdhR*-*aceE*-*aceF*-*lpdA*. Therefore, it is necessary to find the regulatory mechanism of the operon *pdhR*-*aceE*-*aceF*-*lpdA* influenced by FNR and cAMP-CRP.

In summary, we revealed the PdhR-controlled pathways in the present study that support the key regulatory role of PdhR in *P. shigelloides*, laying the groundwork for further investigation of the complex regulatory network of PdhR in bacteria.

## 4. Materials and Methods

### 4.1. Bacterial Strains, Plasmids and Growth Conditions

Appendix A lists the bacterial strains and plasmids used in this study. Bacteria were cultivated at 37 °C (in a shaking incubator) or at 30 °C statically in Luria-Bertani (LB) liquid, solid, and semi-solid medium, M9 medium, and in Dulbecco’s Modified Eagle’s Medium (DMEM) supplemented with 20% fetal bovine serum (FBS). In addition, the anaerobic conditions for the culture of the strains involved in this study were carried out as previously described [52]. When necessary, the media were supplemented with ampicillin (25 μg/mL), chloramphenicol (25 μg/mL), or kanamycin (50 μg/mL).

### 4.2. Genes Deletion and Complementation

In this study, the corresponding gene deletion mutants were constructed with the suicide plasmid pRE112 (sucrose-sensitive lethal) method, which uses the principle of homologous recombination [53]. Briefly, the genomic DNA was used as the template, and the upstream and downstream sequences of the target genes (*pdhR*, *arcA*, *fnr*, and *crp* were the four genes deleted in this study) were amplified by PCR using two pairs of primers F1 and R1, and F2 and R2 (Appendix A), respectively. Furthermore, F1 and R2 contain the restriction sites carried by the pRE112 plasmid, and F2 and R1 contain reverse complementary sequences within 10 bp of each other in their respective 5’ ends. The upstream and downstream sequences were linked together by Overlap-extension PCR using F1 and R2, digested by the above restriction enzymes, and ligated into the pRE112 plasmid before being transferred to *E. coli* S17-1 λpir, which was conjugated with the WT strain. The mutant strains that were successfully homologous recombinant were selected on LB agar plates containing 10% sucrose and the corresponding concentrations of the antibiotics. A schematic illustration of the deletion of all genes in this study is shown in Appendix A. The complementation (Δ*pdhR*/*pdhR*^+^) was generated by using the same method with the corresponding pairs of primers. Agarose gel electrophoresis and DNA sequencing of PCR products were used to confirm the presence of the correct deletion mutations and complementation strains. All primers used in this study are shown in Appendix A.

### 4.3. RNA Isolation and Quantitative Real-Time Polymerase Chain Reaction (qRT-PCR)

To explore the regulatory relationship between regulators and downstream genes, qRT-PCR was performed in the WT and gene deletion mutant strains. The overnight bacterial solution was transferred the following day at a 1:100 ratio to fresh medium at 37 °C with shaking until the bacteria had reached an OD600 of 0.6. Cells were collected by centrifugation at 4 °C, 9000× *g* for 3 min. Total RNA was extracted using TRIzol^®^ Reagent (Invitrogen, Waltham, MA, USA) according to the manufacturer’s protocol, followed by treatment with an RNase-Free DNase. cDNA synthesis was performed using a PrimeScript™ RT reagent Kit (Invitrogen, Waltham, MA, USA). And qRT-PCR analysis was conducted on an Applied Biosystems ABI 7500 sequence detection system (Applied Biosystems, Foster City, CA, USA); meanwhile, the *P. shigelloides gyrB* gene was used as the internal control for qRT-PCR [5], and relative expression levels were calculated as fold change values using the 2^−ΔΔCT^ method [52]. Each experiment was carried out in triplicate.

### 4.4. RNA Sequencing (RNA-Seq)

To identify pathways controlled by PdhR, RNA-Seq of the WT and the *pdhR* deletion strain was carried out for comparison. Briefly, total RNA of the WT and Δ*pdhR* mutant strains was extracted and followed by treatment with an RNase-Free DNase. Subsequently, the RNA was quantified using a NanoDrop 2000 (Thermo Fisher, Waltham, MA, USA), and RNA degradation and contamination were monitored using 1% agarose gels. Following testing and qualification, the cDNA library was sequenced on an Illumina NovaSeq 6000 platform (Illumina, San Diego, CA, USA) to generate 150 bp paired-end reads. Gene expression levels were quantified using HTSeq, and then the Fragments Per Kilobase of transcript sequence per Million base pairs sequenced (FPKM) value of each gene was calculated based on the length of the gene and read count mapped to this gene. The criteria for differentially expressed genes (DEGs) were set as |log2 fold change| ≥ 1 and adjusted *p*-value (padj) ≤ 0.05.

### 4.5. Motility Assay and Transmission Electron Microscopy (TEM) of Flagella

The motility assays were performed as described previously [54]. Briefly, freshly grown WT, Δ*pdhR*, and Δ*pdhR*/*pdhR*^+^ strains were transferred using a sterile toothpick onto swimming agar plates (1% tryptone, 0.5% NaCl, 0.25% agar). The swimming agar plates were incubated at 30 °C for 24 h, and motility was examined by the migration of bacteria through the agar from the center toward the plate periphery. Additionally, TEM and negative staining were used to visualize the flagella of the WT, Δ*pdhR*, and Δ*pdhR*/*pdhR*^+^ strains that were cultured in the swimming agar plates, as previously described [36].

### 4.6. Expression and Purification of Proteins and Electrophoretic Mobility Shift Assays (EMSAs)

The proteins for EMSAs in this study, ArcA and FNR, were cloned into pET-28a, expressed in *E. coli* BL21 (DE3), and purified using a Ni-NTA-Sefinose Column (Sangon Biotech, Shanghai, China). PdhR-maltose-binding protein (MBP) and CRP-Maltose-binding protein fusion proteins were cloned into vector pMAL-c5X, expressed in *E. coli* BL21 (DE3), and purified using amylose resin (New England Bio Labs, Ipswich, MA, USA) affinity chromatography. For EMSAs, according to previous studies [21] with minor modifications, a mixture of each probe (50 ng) and increasing concentrations of PdhR was incubated at 37 °C for 30 min in a 20-μL reaction volume containing 20 mM Tris-HCl (pH 7.5), 100 mM KCl, 10 mM MgCl_2_, 5% glycerol, and 2 mM DTT. Phosphorylation reactions of ArcA (ArcA-P) were carried out as described previously [55], then EMSAs of ArcA-P and each purified promoter fragment were performed by adding increasing concentrations of phosphorylated ArcA-His_6_ fusion proteins in binding buffer (100 mM Tris-HCl (pH 7.5), 10 mM MgCl_2_, 2 mM DTT, 100 mM KCl, 10% glycerol) at 37  °C for 30 min, and non-phosphorylated ArcA (ArcA-P(-)) as the negative control. EMSAs of FNR and each of the purified promoter fragments were performed by adding increasing concentrations of FNR-His_6_ fusion protein in binding buffer (20 mM Tris-HCl (pH 7.5), 80 mM NaCl, 0.1 mM EDTA, 1 mM DTT, and 5% glycerol) at 37  °C for 20 min [56]. Additionally, EMSAs of CRP and each purified promoter fragments in 20 μL of binding buffer (50 mmol Tris-HCl (pH 8.0), 250 mmol/L KCl, 5 mmol/L MgCl_2_, 2.5 mmol/L EDTA, 2.5 mmol/L DTT, 1 μg Poly (dI.dC) and 0.5 μmol/L cAMP) for 30 min at room temperature [57]. Subsequently, DNA-protein complexes were separated by 6% PAGE in 0.5 × TBE buffer at 180 V for 1.5 h. Gels were stained with GelRed for 10 min and imaged using a gel imaging system (GE Healthcare, Chicago, IL, USA).

### 4.7. Chromatin Immunoprecipitation and Quantitative PCR (ChIP-qPCR)

The ChIP assay was performed as previously described with some modifications [58]. Briefly, the pBAD33-*arcA*-3×Flag and pBAD33-*pdhR*-3×Flag recombinant plasmids were electroporated into the Δ*arcA* strains and WT, respectively, and cultured in LB broth containing arabinose at 37 °C, 200 rpm until OD600 reached 0.6. Cross-linking was conducted for 30 min with 1% formaldehyde, followed by 10 min with 0.5 M glycine to quench the cross-linking. Centrifugation was used to separate the bacteria, and PBS was used to wash the pellets three times. The collected bacteria were suspended and incubated in 500 μL lysis buffer (50 mM Tris-HCl (pH 7.5), 100 mM NaCl, 0.5 mg/mL RNase A, 20 mg/mL lysozyme, 1 mM PMSF, and 1 mM EDTA) for 30 min at 37 °C, and then added 500 μL sonication buffer (100 mM Tris-HCl (pH 7.5), 200mM NaCl, 1 mM EDTA, and 2% TritonX-100). The lysate was sonicated with 20 cycles of 5 s on/off at 45% amplitude, and the generated DNA fragments were approximately 200–500 bp. The supernatant was collected by using centrifugation at 4 °C, 15,000× *g* for 10 min, and divided into two equal portions, one of which was added with 20 μL anti-FLAG antibody (Sigma-Aldrich, Shanghai, China), as the ChIP sample, and the other without addition of any antibodies, as the mock-ChIP sample. Both the ChIP and mock-ChIP samples were incubated with the 20 μL protein A magnetic bead (MCE, Plainsboro Township, NJ, USA) at 4 °C for 10 h. The magnetic beads were re-suspended in 20 μL of elution buffer, which contains 50 mM Tris-HCl (pH 8.0), 10 mM EDTA, and 1% SDS, after being cleaned three times with sterile PBS. After elution, the samples were de-cross linked at 65 °C for 5 h, and then 10 μL RNase A (10 mg/mL) was added for RNA decontamination. The eluted DNA samples were further purified, and qRT-PCR was performed to investigate the enrichment folds of the target gene fragment in the ChIP sample relative to the mock-ChIP sample. We conducted the experiments at three time points, with three repetitions for each time point.

### 4.8. Luminescence Screening Assay

The luminescence screening assay was performed as previously described [38]. The amplification products of the respective promoter regions were digested and cloned into the plasmid pMS402. The fusion reporter plasmid was transformed into the relative bacteria and cultured in an LB medium at 37 °C until the mid-logarithmic phase. Promoter activities were measured at OD600 using a Synergy 2 plate reader (Agilent BioTek, Santa Clara, CA, USA). Each experiment was carried out in triplicate.

### 4.9. Growth Assay

The dynamic growth experiment for the WT, Δ*pdhR,* and Δ*pdhR*/*pdhR*^+^ strains was carried out in LB medium, M9 medium, and DMEM, and the experiment procedure and data analysis were conducted completely as described in a previous study [59]. Briefly, the bacterial strains were grown in sterile media at 37 °C overnight with shaking. The overnight bacterial solution was transferred the following day at a 1:100 ratio to a fresh medium until the bacteria had reached an OD600 of 0.6. Then, at a ratio of 1:200 per well, the bacterial solution was added to five wells of a 96-well cell plate containing 200 μL of medium at a ratio of 1:200 per well. As a control, fresh medium was added to the nearby wells. Finally, for the dynamic growth experiment, the prepared 96-well cell plate was placed in a Molecular Devices Spectra MAX 190 full-wavelength microplate reader (Molecular Devices, San Jose, CA, USA) to be carried out. Each experiment was carried out in triplicate.

### 4.10. Invasion Assay

The invasion assay was carried out as previously described, with some modifications [60]. Briefly, the WT, Δ*pdhR,* and Δ*pdhR*/*pdhR*^+^ strains were grown overnight in LB, and transferred to fresh LB at an inoculation ratio of 1:100 until the bacteria were grown to OD_600_ = 0.6. The bacterial cells were then pelleted by centrifugation, and the supernatant was discarded. Approximately 5 × 10^7^ WT, Δ*pdhR,* and Δ*pdhR*/*pdhR*^+^ bacterial cells were layered onto confluent monolayers of approximately 1 × 10^5^ Caco-2 cells per well in 24-well plates. Then, Caco-2 cells and bacterial cells were co-cultured at 37 °C in 5% CO_2_ for 1 h to initiate invasion. Subsequently, 100 μg/mL gentamicin was added to the cell culture medium 1 h post-infection, and the cells were incubated for 40 min to kill the extracellular bacteria. After incubation, the cells were washed with PBS and lysed using 0.15% Triton X-100 for 10 min to release the intracellular bacterial cells. The invasion rate was calculated as the ratio of the number of recovered bacteria to the total number of bacterial cells used for infection. We conducted the assay at three time points, with six repetitions for each time.

### 4.11. Statistical Analysis

Data were analyzed using GraphPad Prism v7.0 software (GraphPad Inc., La Jolla, CA, USA) [61]. All data are expressed as means ± standard deviation (SD). Student’s *t*-test was used to analyze significant differences between the two groups. A probability value (*p*) ≤ 0.05 was considered statistically significant (in the figures, *** *p* ≤ 0.001; ** *p* ≤ 0.01; * *p* ≤ 0.05; NS indicates not significant). Construction of the PdhR evolutionary tree used the Molecular Evolutionary Genetics Analysis (MEGA) software version 6.0 [62].

## Figures and Tables

**Figure 1 ijms-24-14473-f001:**
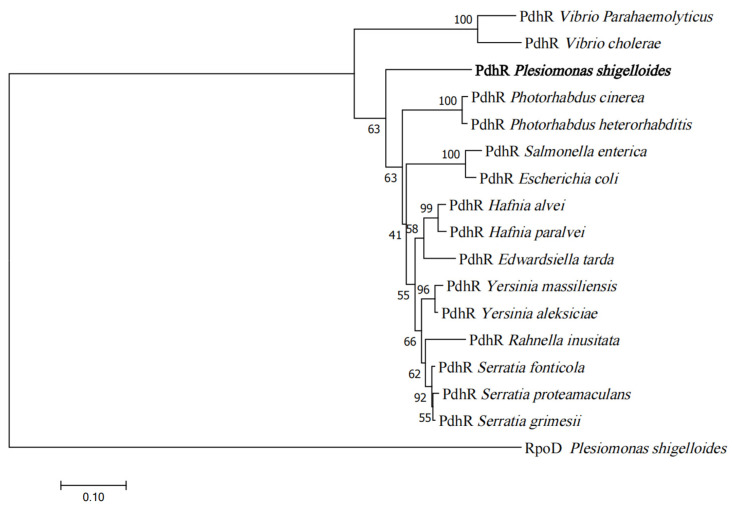
Phylogenetic analysis of PdhR. An unrooted phylogenetic tree constructed using the neighbor-joining method based on PdhR amino acid sequences; bootstrap values were based on 1000 replications. All amino acid sequences were downloaded from the National Center for Biotechnology Information. The numbers in the figure represented the homologous similarity of PdhR in different strains, while PdhR in *P. shigelloides* was in bold font.

**Figure 2 ijms-24-14473-f002:**
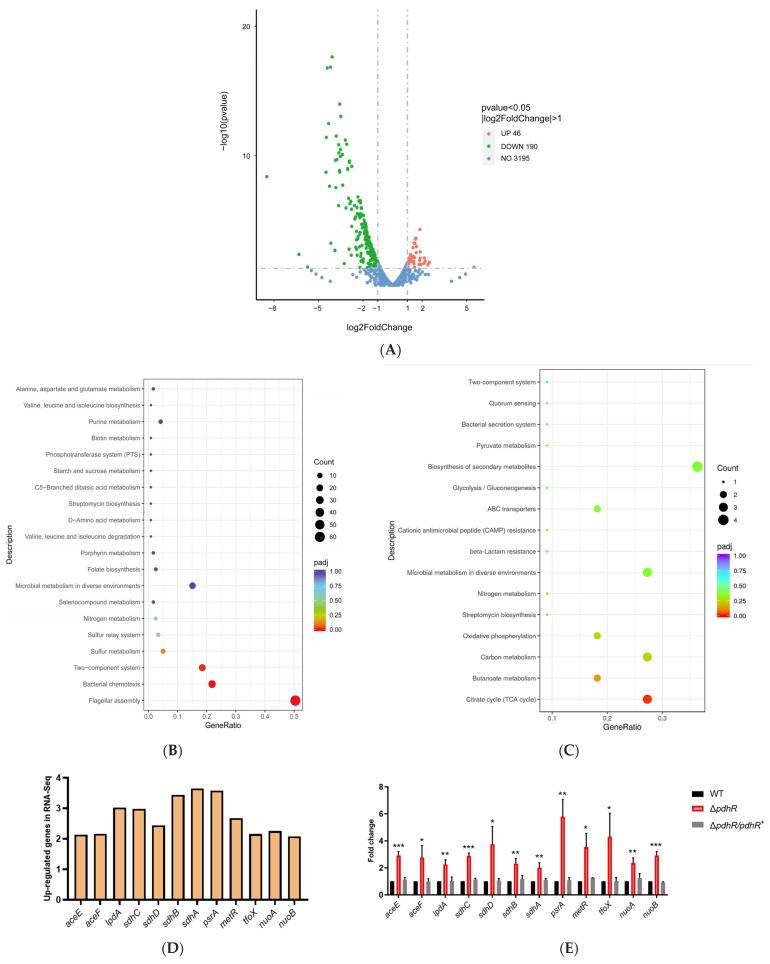
Transcriptomic analysis of *P. shigelloides* between WT and Δ*pdhR* strains. (**A**) The volcano plot of differentially expressed genes (DEGs); the red circle indicates up-regulated genes, the green circle indicates down-regulated genes, and the blue circle indicates no DEGs. (**B**) KEGG enrichment of down-regulated DEGs. (**C**) KEGG enrichment of up-regulated DEGs; the GeneRatio refers to the ratio of the number of DEGs in the pathway and the number of all annotated genes in the pathway. (**D**) Twelve upregulated DEGs in the transcriptome profiles were selected for validation using qRT-PCR in the WT, Δ*pdhR*, and Δ*pdhR*/*pdhR*^+^ strains (**E**). (**F**) Thirteen downregulated DEGs in the transcriptome profiles were selected for validation using qRT-PCR in the WT, Δ*pdhR*, and Δ*pdhR*/*pdhR*^+^ strains (**G**) (*** *p* ≤ 0.001; ** *p* ≤ 0.01; * *p* ≤ 0.05).

**Figure 3 ijms-24-14473-f003:**
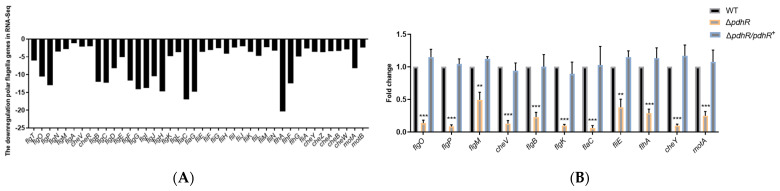
PdhR influences motility and flagellar synthesis by positively regulating the expression of flagellar genes in *P. shigelloides.* (**A**) The transcription levels of polar flagellar genes in RNA-seq. (**B**) qRT-PCR was performed to validate the significant downregulation of 11 polar flagellar genes in RNA-seq. (**C**) The transcription levels of lateral flagellar genes in RNA-seq. (**D**) qRT-PCR was performed to validate the significant downregulation of nine lateral flagellar genes in RNA-seq. (**E**) Four polar and five lateral flagellar genes were chosen to construct promoter-lux fusions in the Δ*pdhR* mutant and WT strains to confirm the results of qRT-PCR and RNA-seq. (**F**) The motility of WT, Δ*pdhR*, and Δ*pdhR*/*pdhR*^+^ strains grown in swimming agar plate. (**G**) TEM visualization of the flagella produced by the WT, Δ*pdhR*, and Δ*pdhR*/*pdhR*^+^ strains. The hollow bacterial flagella were pointed by the colored arrows (*** *p* ≤ 0.001; ** *p* ≤ 0.01; * *p* ≤ 0.05).

**Figure 4 ijms-24-14473-f004:**
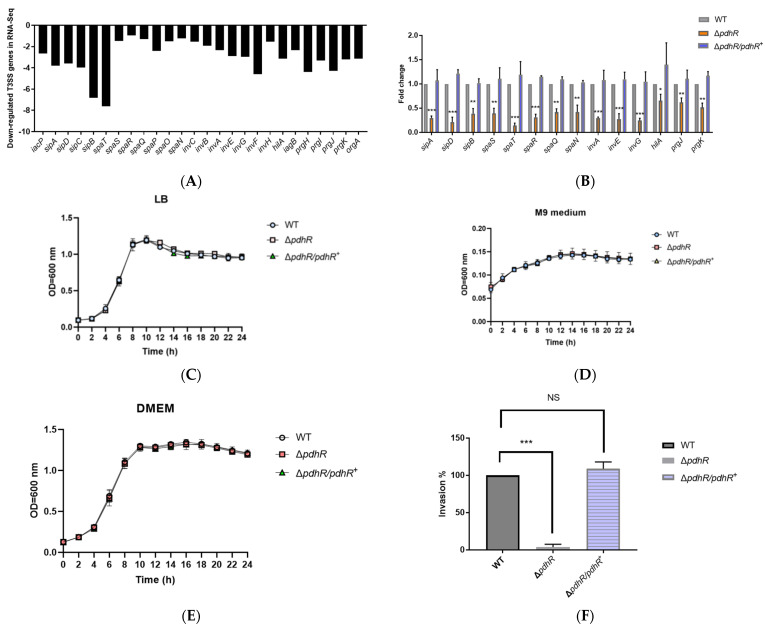
PdhR promotes *P. shigelloides*’ ability to infect Caco-2 cells by positively regulating T3SS expression. (**A**) The transcription levels of T3SS cluster in RNA-seq. (**B**) qRT-PCR was performed to validate 14 virulence genes from the T3SS cluster in RNA-seq. (**C**) The dynamic growth experiment for the WT, Δ*pdhR,* and Δ*pdhR*/*pdhR*^+^ strains was carried out in LB medium, (**D**) M9 medium, and (**E**) DMEM. (**F**) Invasion assay of the WT, Δ*pdhR,* and Δ*pdhR*/*pdhR*^+^ strains (*** *p* ≤ 0.001; ** *p* ≤ 0.01; * *p* ≤ 0.05).

**Figure 5 ijms-24-14473-f005:**
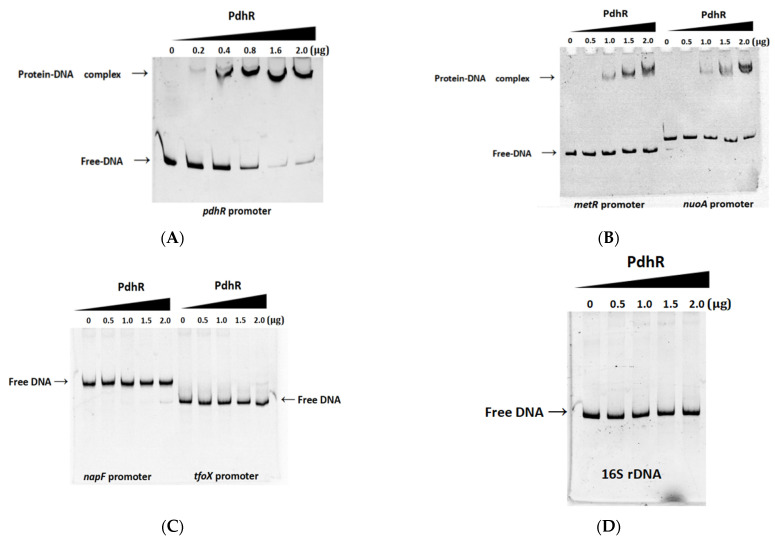
Pyruvate-sensing PdhR directly negatively regulates the expression of *pdhR-aceEF*, *metR,* and *nuoA*. (**A**) The EMSAs between the PdhR protein and the *pdhR* promoter. (**B**) The EMSAs between the PdhR protein, and the *metR* and *nuoA* promoter. (**C**) The EMSAs between PdhR protein, and the *napF* and *tfox* promoter. (**D**) The EMSAs between PdhR protein and 16S rDNA. (**E**) The expression levels of *pdhR*, *metR,* and *nuoA* promoter–lux fusions in the WT and Δ*pdhR* strains. (**F**) the PdhR proteins were enriched at the *pdhR*, *metR,* and *nuoA* promoters in the ChIP samples. (**G**) The expression levels of the operon *pdhR*-*aceE*-*aceF*-*lpdA*, *metR*, and *nuoA* on either LB medium or (**H**) M9 medium with 0.1% pyruvate by qRT-PCR (** *p* ≤ 0.01; * *p* ≤ 0.05).

**Figure 6 ijms-24-14473-f006:**
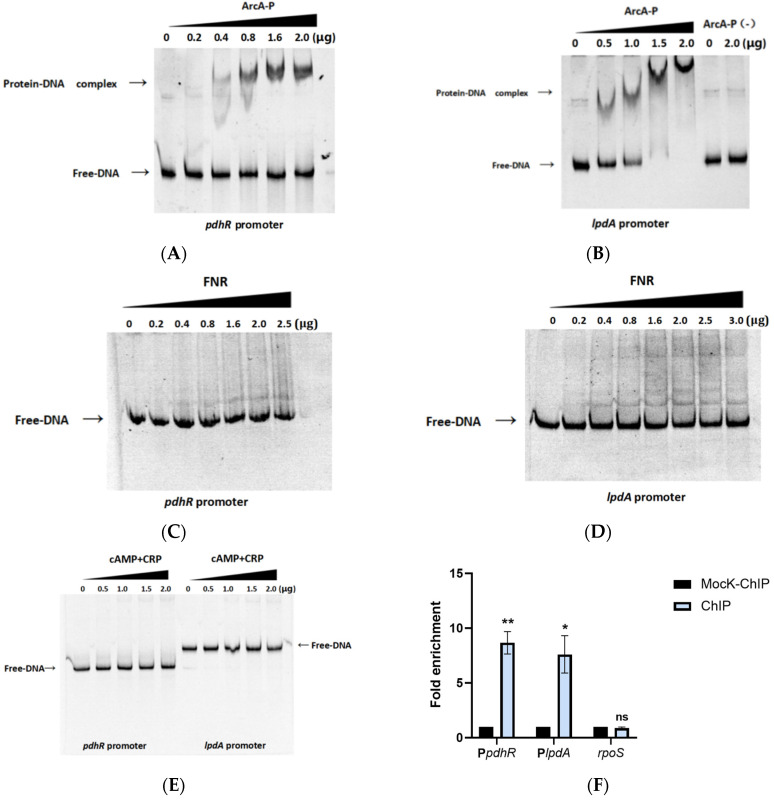
*pdhR* is directly negatively regulated by ArcA and indirectly positively regulated by FNR and CRP. (**A**) The EMSAs between phosphorylated ArcA protein and the *pdhR* promoter. (**B**) The EMSAs between phosphorylated ArcA, unphosphorylated ArcA, and the *lpdA* promoter. (**C**) The EMSAs between FNR and the *pdhR* promoter. (**D**) The EMSAs between FNR and the *lpdA* promoter. (**E**) The EMSAs between the cAMP-CRP protein complex, and the *pdhR* and *lpdA* promoter. (**F**) The ArcA proteins were enriched at the *pdhR* and *lpdA* promoters in the ChIP samples. (**G**) The expression levels of the operon *pdhR*-*aceE*-*aceF*-*lpdA* under anaerobic conditions in the Δ*arcA*, Δ*fnr,* and WT strains by qRT-PCR. (**H**) The expression levels of *pdhR* and *lpdA* promoter-lux fusions in the WT, Δ*arcA,* and Δ*fnr* under anaerobic conditions. (**I**) The expression levels of the operon *pdhR*-*aceE*-*aceF*-*lpdA* in the Δ*crp* and WT strains by qRT-PCR. (**J**) The expression levels of *pdhR* and *lpdA* promoter-lux fusions in the WT and Δ*crp* (*** *p* ≤ 0.001; ** *p* ≤ 0.01; * *p* ≤ 0.05).

**Figure 7 ijms-24-14473-f007:**
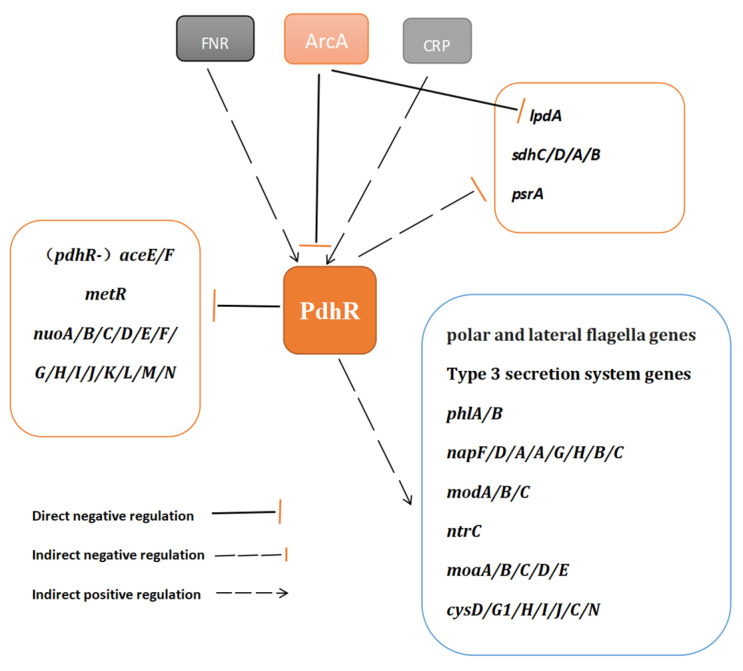
Schematic of the proposed PdhR regulatory mechanism in *P. shigelloides*. The potential regulatory pathways and interplays of PdhR are proposed according to our observations.

## Data Availability

All data are presented within the manuscript and the Appendix A.

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
