# Peer review of "Transcriptome Analysis Reveals the Effect of PdhR in *Plesiomonas shigelloides"

_ijms, 2023, doi:10.3390/ijms241914473_

Round 1
Reviewer 1 Report
The manuscript presents a detailed investigation into the role and significance of PdhR in Plesiomonas shigelloides. The topic is of clear relevance, given the importance of understanding the regulatory mechanisms of opportunistic pathogens.
The introduction provides a helpful insight into the significance of the research, although a more comprehensive examination of the existing literature could improve the quality of the work.
A short and comprehensive review of the most recent collection of research concerning PdhR, with a particular focus on its role across closely associated bacterial species, would provide a solid basis for the manuscript.
In the results section, the authors presented the phylogenetic analysis of PdhR and its conservation across diverse bacterial species. However, it would be advantageous to provide a more comprehensive discussion of the impact of these findings. One aspect that needs further exploration is the relative importance of PdhR concerning specific bacterial species, as it appears to share a closer association with certain species than others.
Transcriptome profiling provides a vast amount of data; however, there is a need for a more accurate explanation of the effects associated with the identification of differentially expressed genes. The manuscript's impact could be enhanced by establishing explicit links between the reported genetic changes and their potential physiological or pathological implications.
In the discussion section, It's essential to draw parallels with existing research. How do the study's findings align with or diverge from the current understanding of PdhR's role? Any discrepancies or novel findings should be highlighted and discussed in depth.
Moreover, it is crucial to highlight the broader implications of understanding the regulatory mechanisms of PdhR.
How might these findings influence disease treatment or prevention strategies?
The manuscript should ensure consistent terminology, especially when discussing genes and proteins. i.e., genes (e.g., pdhR, napF, metR, nuoA, tfoX) and proteins (e.g., PdhR, FNR, ArcA, cAMP-CRP) names should be consistent terminology.
Author Response
Reply Letter
Manuscript ID: ijms-2609506
Manuscript Title: Transcriptome analysis reveals the effect of PdhR in Plesiomonas shigelloides
Dear reviewer
Thank you very much for your valuable efforts in handling this manuscript, ijms-2609506. We are also very grateful to you for your constructive comments, which helped us improve the quality of this research paper.
We hereby resubmit a revised manuscript to IJMS after making revisions and carefully considering all the comments. On the following pages, the detailed point-by-point responses to each comment are provided.
Thank you very much for your continued consideration of our work.
Best Regards,
Boyang Cao, Professor
Nankai University
Overview from reviewer 1:
The manuscript presents a detailed investigation into the role and significance of PdhR in Plesiomonas shigelloides. The topic is of clear relevance, given the importance of understanding the regulatory mechanisms of opportunistic pathogens.
Minor Comments:
1 The introduction provides a helpful insight into the significance of the research, although a more comprehensive examination of the existing literature could improve the quality of the work.
Our response: Thank you for the suggestions. The relevant explanation was added in page 7, lines 155-159 of the revised manuscript.
2 A short and comprehensive review of the most recent collection of research concerning PdhR, with a particular focus on its role across closely associated bacterial species, would provide a solid basis for the manuscript.
Our response: Thanks for your suggestion. At present, almost all the reports on PdhR come from the study of E. coli, and there are few reports on other strains. We have summarized the relevant content as far as possible in the introduction section, and the relevant explanation was added in pages 6, lines 130–132 of the revised manuscript. Meanwhile, we sincerely hope that our work can provide certain reference value and significance for the study of PdhR in other bacterial species.
3 In the results section, the authors presented the phylogenetic analysis of PdhR and its conservation across diverse bacterial species. However, it would be advantageous to provide a more comprehensive discussion of the impact of these findings. One aspect that needs further exploration is the relative importance of PdhR concerning specific bacterial species, as it appears to share a closer association with certain species than others.
Our response: Thanks for your suggestion. The relevant explanation was added in page 8, lines 169-179 of the revised manuscript.
4 Transcriptome profiling provides a vast amount of data; however, there is a need for a more accurate explanation of the effects associated with the identification of differentially expressed genes. The manuscript's impact could be enhanced by establishing explicit links between the reported genetic changes and their potential physiological or pathological implications.
Our response: Thanks for your suggestion. The relevant explanation was added in page 10, lines 205-215 of the revised manuscript.
5 In the discussion section, It's essential to draw parallels with existing research. How do the study's findings align with or diverge from the current understanding of PdhR's role? Any discrepancies or novel findings should be highlighted and discussed in depth.
Our response: Thanks for your suggestion. The relevant explanation was added in page 24, lines 530-536, page 24, lines 544-548 of the revised manuscript.
6 Moreover, it is crucial to highlight the broader implications of understanding the regulatory mechanisms of PdhR.
Our response: Thanks for your suggestion. The relevant explanation was added in page 27, lines 593-596 of the revised manuscript.
7 How might these findings influence disease treatment or prevention strategies?
Our response: In this study, we found that PdhR positively regulates flagella synthesis and T3SS, thereby affecting the motility and virulence of P. shigelloides. Gram-negative bacteria are known to use a wide range of virulence factors to subvert eukaryotic cell physiological systems, with the type three secretion system (T3SS) being one of the most important. The T3SS is a needle-like device that the bacterium employs to inject a varied collection of effector proteins straight into the cytoplasm of host cells, where they can disrupt the host cellular machinery for a variety of reasons [1]. Furthermore, flagella is a virulence factor for many bacteria, and we also found that PdhR positively regulates hemolysin (phlA/B). The above-mentioned are basically essential for pathogenic bacteria to invade the host and cause the disease; they can be used as a future research direction for the prevention or treatment of P. shigelloides infection. At the same time, we will further explore the more comprehensive and specific pathogenic mechanism of P. shigelloides in future studies.
[1] Rahmatelahi H, El-Matbouli M, Menanteau-Ledouble S. Delivering the pain: an overview of the type III secretion system with special consideration for aquatic pathogens. Vet Res. 2021 Dec 19;52(1):146.
8 The manuscript should ensure consistent terminology, especially when discussing genes and proteins. i.e., genes (e.g., pdhR, napF, metR, nuoA, tfoX) and proteins (e.g., PdhR, FNR, ArcA, cAMP-CRP) names should be consistent terminology.
Our response: Thanks for your suggestion. We carefully examined the manuscript and modified that genes (e.g., pdhR, napF, metR, nuoA, tfoX) and proteins (e.g., PdhR, FNR, ArcA, cAMP-CRP) names stated correctly and consistently throughout the text.

Reviewer 2 Report
The article titled " Transcriptome analysis reveals the effect of PdhR in Plesiomonas shigelloides" by Yang et al,,presents a comprehensive exploration of the pyruvate dehydrogenase complex regulator (PdhR) in Plesiomonas shigelloides and its impact on various cellular processes.
The study adopts RNA sequencing (RNA-Seq) to compare wild-type (WT) and ΔpdhR mutant strains, revealing that PdhR governs a significant portion, approximately 7.38%, of the P. shigelloides transcriptome. One of the key findings of this research is the identification of PdhR's influence on flagellar genes, resulting in the downregulation of both polar and lateral flagella genes in the ΔpdhR mutant, which consequently led to a non-motile phenotype. This observation was corroborated by motility assays and transmission electron microscopy (TEM), demonstrating the absence of flagella.Moreover, the study illuminates the role of PdhR in the regulation of the Type III Secretion System (T3SS) cluster. The ΔpdhR mutant exhibited reduced infectivity towards Caco-2 cells compared to the WT strain, emphasizing the significance of PdhR in P. shigelloides pathogenicity.The article further delves into the molecular mechanisms underlying PdhR's regulatory functions. It confirms that PdhR directly binds to its promoter, suppressing the expression of the pdhR-aceEF-lpd operon. Additionally, the study identifies two downstream genes, metR and nuoA, as being negatively regulated by PdhR. Furthermore, the research highlights the presence of ArcA upstream of pdhR and lpdA, revealing ArcA's role in negatively regulating their expression.
Overall, this article offers valuable insights into the multifaceted functions of PdhR in Plesiomonas shigelloides, shedding light on its regulatory pathways and their implications for pathogenicity. Before the manuscript is accepted,I suggest the authors please modify or explain below comments:
- Figure one was not in the main text. It would be whether this figure is combined with figure 2 or supplemental.
- Please modify Figure 2 especially since the graphs and letters were very blurry.
- Why in the qRT-PCR graphs especially in the WT sample there were no error bars? Could the author explain this?
- In TEM and motility figure. Polar and lateral flagella were not differentiated. Could the author explain how they perform swimming assay? What was the agar concentration used? Whether TEM experiments were performed in liquid culture or solid medium.
- It would be good if you added any reference in the discussion section in which polar or lateral flagella were regulated by negative regulation in any other system.
Author Response
Reply Letter
Manuscript ID: ijms-2609506
Manuscript Title: Transcriptome analysis reveals the effect of PdhR in Plesiomonas shigelloides
Dear reviewer
Thank you very much for your valuable efforts in handling this manuscript, ijms-2609506. We are also very grateful to you for your constructive comments, which helped us improve the quality of this research paper.
We hereby resubmit a revised manuscript to IJMS after making revisions and carefully considering all the comments. On the following pages, the detailed point-by-point responses to each comment are provided.
Thank you very much for your continued consideration of our work.
Best Regards,
Boyang Cao, Professor
Nankai University
Comments for Authors from reviewer 2 :
The article titled " Transcriptome analysis reveals the effect of PdhR in Plesiomonas shigelloides" by Yang et al,, presents a comprehensive exploration of the pyruvate dehydrogenase complex regulator (PdhR) in Plesiomonas shigelloides and its impact on various cellular processes.
The study adopts RNA sequencing (RNA-Seq) to compare wild-type (WT) and ΔpdhR mutant strains, revealing that PdhR governs a significant portion, approximately 7.38%, of the P. shigelloides transcriptome. One of the key findings of this research is the identification of PdhR's influence on flagellar genes, resulting in the downregulation of both polar and lateral flagella genes in the ΔpdhR mutant, which consequently led to a non-motile phenotype. This observation was corroborated by motility assays and transmission electron microscopy (TEM), demonstrating the absence of flagella. Moreover, the study illuminates the role of PdhR in the regulation of the Type III Secretion System (T3SS) cluster. The ΔpdhR mutant exhibited reduced infectivity towards Caco-2 cells compared to the WT strain, emphasizing the significance of PdhR in P. shigelloides pathogenicity. The article further delves into the molecular mechanisms underlying PdhR's regulatory functions. It confirms that PdhR directly binds to its promoter, suppressing the expression of the pdhR-aceEF-lpd operon. Additionally, the study identifies two downstream genes, metR and nuoA, as being negatively regulated by PdhR. Furthermore, the research highlights the presence of ArcA upstream of pdhR and lpdA, revealing ArcA's role in negatively regulating their expression.
Overall, this article offers valuable insights into the multifaceted functions of PdhR in Plesiomonas shigelloides, shedding light on its regulatory pathways and their implications for pathogenicity. Before the manuscript is accepted, I suggest the authors please modify or explain below comments:
Minor Comments:
1 Figure one was not in the main text. It would be whether this figure is combined with figure 2 or supplemental.
Our response: Thanks for your advice. Figure 1 was reupload in page 9, lines 183-191 of the revised manuscript.
2 Please modify Figure 2 especially since the graphs and letters were very blurry.
Our response: Thank you for your attention. Modified.
3 Why in the qRT-PCR graphs especially in the WT sample there were no error bars? Could the author explain this?
Our response: Thanks for your suggestion. The expression level of the corresponding genes was tested at least three times when performing the qRT-PCR experiments, so there were error bars. Well, for the WT, it's normalized to 1 for each time, so there was no error bars.
4 In TEM and motility figure. Polar and lateral flagella were not differentiated. Could the author explain how they perform swimming assay? What was the agar concentration used? Whether TEM experiments were performed in liquid culture or solid medium.
Our response: Thank you for your attention. P. shigelloides is the unique member of the Enterobacteriaceae family that is able to produce polar and lateral flagella [a]. Bacteria that possess functional dual flagella systems that are able to express both a constitutive polar flagellum required for swimming motility and a separate lateral flagella system that is induced in viscous media or on surfaces and is essential for swarming motility [b]. In this study, we discovered that PdhR positively regulates a significant number of flagellar genes using transcriptome analysis, and we subsequently focused on the effects of PdhR on the motility capacity and flagellar synthesis of P. shigelloides. And we will seriously consider your suggestions and try to distinguish the polar and lateral flagella formed by P. shigelloides in subsequent studies.
The swimming agar plates used 0.25% agar for motility assays, and the bacteria cultured in this semisolid medium were selected for TEM experiments. The relevant explanation was added in page 31, line 714 and page 32, line 718 of the revised manuscript.
[a] Inoue K, Kosako Y, Suzuki K, Shimada T. Peritrichous flagellation in Plesiomonas shigelloides strains. Jpn J Med Sci Biol. 1991;44(3):141-6.
[b] Merino S, Shaw JG, Tomás JM. Bacterial lateral flagella: an inducible flagella system. FEMS Microbiol Lett. 2006 Oct;263(2):127-35.
5 It would be good if you added any reference in the discussion section in which polar or lateral flagella were regulated by negative regulation in any other system.
Our response: Thank you for your suggestion. We found less coherence in the front and back content after adding the content, which polar or lateral flagella were governed by negative regulation in any other system, to the corresponding location in the discussion section. So we included this section in the response, which you are welcome to browse. Thank you again for your recognition and advice on our research.
There are currently few reports on the simultaneous regulation of both flagella gene clusters by regulators in strains that possess polar and lateral flagella. In addition to our finding that PdhR can simultaneously positively regulate the expression of both flagella gene clusters in P. shigelloides, there are also studies claiming that LuxS negatively regulates the expression of polar and lateral flagella genes in Vibrio harveyi as well as swimming and swarming abilities [c].
[c] Zhang YQ, Deng YQ, Feng J, Hu JM, Chen HX, Guo ZX, Su YL. LuxS modulates motility and secretion of extracellular protease in fish pathogen Vibrio harveyi. Can J Microbiol. 2022 Mar;68(3):215-226.
